# The Clinical Role of Adjuvant Chemotherapy after Sublobar Resection for Non-Small-Cell Lung Cancer ≤ 20 mm with Lymph Node Metastases: A Propensity-Matched Analysis of the National Cancer Database

**DOI:** 10.3390/cancers16122176

**Published:** 2024-06-08

**Authors:** Shinkichi Takamori, Junjia Zhu, Asato Hashinokuchi, Takefumi Komiya

**Affiliations:** 1Department of Thoracic and Breast Surgery, Oita University Faculty of Medicine, Oita 870-1124, Japan; s-takamori@oita-u.ac.jp; 2Department of Public Health Sciences, Penn State College of Medicine, Hershey, PA 17033, USA; jzhu2@pennstatehealth.psu.edu; 3Department of Surgery and Science, Graduate School of Medical Sciences, Kyushu University, Fukuoka 819-0395, Japan; hashinokuchi.asato.444@m.kyushu-u.ac.jp; 4Division of Hematology Oncology, Penn State College of Medicine, Hershey, PA 17033, USA

**Keywords:** non-small-cell lung cancer, surgery, adjuvant chemotherapy, sublobar resection, lymph node metastasis

## Abstract

**Simple Summary:**

Sublobar resection is commonly used for treating small-sized non-small-cell lung cancer (NSCLC) patients. The effectiveness of adjuvant chemotherapy for these patients, particularly those with pathological lymph node metastasis, remains unclear. A study using data from the National Cancer Database from 2004 to 2018 reviewed cases of NSCLC patients who had undergone sublobar resection with lymph node metastasis and met specific criteria, including tumor size ≤ 20 mm and no comorbidities. The study found that, after adjusting for age, sex, histologic type, and pathological lymph node status through propensity score matching, patients who received adjuvant chemotherapy had a notably longer survival than those who did not (median survival of 64.3 vs. 34.0 months). The results indicate that adjuvant chemotherapy significantly enhances survival in small-sized-NSCLC patients with lymph node involvement post-sublobar resection.

**Abstract:**

Sublobar resection is a standard surgical procedure for small-sized non-small-cell lung cancer (NSCLC). However, the clinical role of adjuvant chemotherapy for small-sized NSCLC with pathological lymph node (LN) metastasis after sublobar resection is unknown. The National Cancer Database was queried for NSCLC patients between 2004 and 2018. Eligibility included sublobar resection with pathological LN metastasis, R0 resection, Charlson comorbidity score = 0, clinical stage T1a-b, and tumor size ≤ 20 mm. The Kaplan–Meier method with a log-rank test and multivariable Cox proportional hazards analyses were used for assessing survival. The samples were evaluated before and after propensity score matching (PSM) with respect to age, sex, histologic type, and pathological LN status. Of 810 patients who met the eligibility criteria, 567 (70.0%) underwent adjuvant chemotherapy. After PSM, patients with adjuvant chemotherapy had a significantly longer survival than those without (median survival: 64.3 vs. 34.0 months, hazard ratio for death: 0.61, *p* < 0.0001). Multivariate analyses after PSM showed that younger age (*p* = 0.0206), female (*p* = 0.0005), and adjuvant chemotherapy (*p* < 0.0001) were independent prognostic factors for longer survival. Adjuvant chemotherapy has a prognostic impact in patients with small-sized NSCLC and pathological lymph node metastasis who undergo sublobar resection.

## 1. Introduction

Lung cancer remains the primary cause of death from cancer [1]. Non-small-cell lung cancer (NSCLC) accounts for most cases of lung cancer. Although lobectomy and lymphadenectomy have been the standard treatment for NSCLC patients with early-stage disease [2], more than 50% patients relapse after surgery alone [3]. Previous clinical trials investigating patients with stage I–III NSCLC post-surgery have shown that cisplatin-based adjuvant therapy significantly lowers mortality risk, particularly in stages II and III diseases [4]. The efficacy of cytotoxic therapy is grounded in an intensive cisplatin-based regimen to eradicate micrometastatic lesions [5].

In Japan, the use of chest computed tomography for lung cancer screening has led to the increased detection of early-stage lung cancers. Specifically, it has become evident that NSCLC with a maximum tumor diameter of 2 cm or less, located in the lung periphery, has an exceptionally favorable prognosis [6,7,8]. An analysis of 6644 patients who underwent resection of the aforementioned NSCLC included a subgroup analysis where clinical stage IA was divided into tumors measuring 2 cm or less and those measuring 2–3 cm. The 5-year survival rates were 77.5% and 69.3%, respectively, with a statistically significant difference favoring the former group [9]. These results indicate that, clinically, the former and latter groups are recognized as populations with different prognoses [10,11,12,13]. In recent years, several clinical trials have confirmed the non-inferior efficacy of sublobar resection compared to lobectomy in small-sized NSCLC. In patients diagnosed with peripheral NSCLC with tumors up to 2 cm in size and verified absence of nodal involvement in the hilar and mediastinal regions, sublobar resection demonstrated comparable outcomes to lobectomy in terms of disease-free survival (DFS) [14]. In addition, a recent clinical trial in Japan comparing lobectomy and segmentectomy for small-sized (≤20 mm) peripheral NSCLC revealed that DFS for segmentectomy was not inferior to lobectomy, and overall survival (OS) was even significantly longer in the segmentectomy group compared to lobectomy [15]. As the number of patients with early-stage NSCLC suitable for sublobar resection is expected to increase, necessitating increased focus on the clinical role of adjuvant chemotherapy for these patients with pathological lymph node (LN) metastasis. Patients with NSCLC who have undergone lobectomy are indicated for postoperative adjuvant chemotherapy if LN metastases are found postoperatively. Strictly speaking, however, there is no evidence for the efficacy of adjuvant chemotherapy after sublobar resection of small-sized NSCLC. Thus, the aim of the current study was to examine the prognostic significance of adjuvant chemotherapy in patients with small-sized (≤20 mm) NSCLC and pathological LN metastasis who underwent sublobar resection. A graphical abstract is shown in Figure 1.

## 2. Materials and Methods

### 2.1. National Cancer Database (NCDB)

The NCDB represents a collaborative initiative between the Commission on Cancer (CoC) at the American College of Surgeons and the American Cancer Society. The data utilized in this study, provided by hospitals participating in the CoC’s NCDB, are de-identified, and thus, the institutions mentioned do not confirm the accuracy or endorse the conclusions of the data analysis presented by the authors. This dataset is considered hospital-based rather than population-based. The Penn State College of Medicine Institutional Review Board granted an exemption for this study. The process for selecting eligible cases is shown in (Figure 2). The purpose of the current study was to investigate the survival benefit of adjuvant chemotherapy in patients with small-sized (≤2 cm) NSCLC who underwent “intentional sublobar resection”, even if they were eligible for lobectomy. Thus, the inclusion criteria included clinical stage IB or less, comorbidity score = 0, and tumor size ≤ 2 cm to exclude “non-intentional sublobar resection”. Patients diagnosed with pathological N1 or N2 NSCLC and captured in the NCDB database between 2004 and 2018 were selected (*n* = 89,438). Of these, patients with pathological M1 disease were excluded (*n* = 4220). Patients with R0 resection (*n* = 74,552), Charlson comorbidity score = 0 (*n* = 65,819), tumor size ≤ 20 mm (*n* = 5878), clinical T1b or less disease (*n* = 5222), and sublobar resection (*n* = 810) were eligible for final analysis. The number of LN dissections was classified into 3 groups (≥10, ≤9, and unknown) according to the CoC’s recommendation [16]. Regarding the types of chemotherapy used, given that adjuvant atezolizumab and pembrolizumab were approved by the FDA in 2021 and 2023, respectively, the majority of the types of adjuvant chemotherapy used in this study would have been platinum-based, and very few cases of immune checkpoint inhibitors, if any, being used in a clinical trial setting were included [17,18].

### 2.2. Statistical Analysis

Patients’ demographics and basic clinical characteristics were compared between the adjuvant chemotherapy groups using chi-square tests. Overall survival (OS) was defined as the time (years) from diagnosis to death from any cause. Kaplan–Meier curves according to the use of adjuvant chemotherapy were compared using the log-rank test. Univariate and multivariable Cox proportional hazards analyses were performed using JMP^®^ 14.0 (SAS Institute Inc., Cary, NC, USA). Analyses using propensity score matching (PSM) were conducted with the aim of reducing the bias of the observational study. The propensity scores included matching of the following variables: age, sex, histology, and pathological N stage. PSM was performed using the “MatchIt” package version 4.5.5 running on R version 4.3.3 (R Foundation for Statistical Computing, Vienna, Austria), and 1:1 nearest-neighbor matching was performed with propensity scores estimated through logistic regression. Finally, 243 matched patients from each group were included in the PSM analyses. All tests were two-sided and *p* < 0.05 was considered statistically significant.

## 3. Results

### 3.1. Patient Characteristics

The patient characteristics (*n* = 810) are summarized in Table 1. In total, 567 (70%) received adjuvant chemotherapy, 651 (80%) underwent wedge resection, 159 (20%) underwent segmentectomy, and 526 (65%) had pathological N2 disease. Patients who received adjuvant chemotherapy were significantly associated with younger age (*p* < 0.0001), non-academic institution (*p* = 0.0060), late years of diagnosis (*p* = 0.0026), right laterality (*p* = 0.0241), wedge resection (*p* = 0.0058), pathologic N2 stage (*p* < 0.0001), and adjuvant chest radiation therapy (*p* < 0.0001). After PSM adjusting age, sex, histology, and pathological N status, these factors were well balanced between two groups with or without adjuvant chemotherapy.

### 3.2. Univariate Analyses of OS in NSCLC Patients with Pathological LN Metastasis Who Underwent Sublobar Resection According to the Use of Adjuvant Chemotherapy

The Kaplan–Meier curves comparing OS according to the use of adjuvant chemotherapy in patients who underwent sublobar resection with pathological LN metastasis are shown in Figure 3. Patients with adjuvant chemotherapy had a significantly longer OS than those without (5-year survival rate: 50.3% vs. 34.2%, median OS: 62.0 vs. 34.0 months, hazard ratio [HR] for death: 0.62, 95% confidence interval [CI]: 0.52–0.74, *p* < 0.0001; Figure 3a). After PSM for age, sex, histology, and pathological LN metastasis, patients with adjuvant chemotherapy had a significantly longer OS than those without (5-year survival rate: 50.6% vs. 34.2%, median OS: 64.3 vs. 34.0 months, HR for death: 0.61, 95% CI: 0.49–0.75, *p* < 0.0001; Figure 3b). The 5-year survival benefit by adjuvant chemotherapy was 16.4%.

### 3.3. Univariate and Multivariable Analyses of OS in NSCLC Patients with Pathological LN Metastasis Who Underwent Sublobar Resection

The results of univariate and multivariable analyses for OS in small-sized NSCLC patients who underwent sublobar resection are shown in Table 2. Univariate analysis showed that younger age (*p* = 0.0026), female sex (*p* < 0.0001), number of LNs dissected ≥10 (*p* = 0.0073), pathological N1 disease (*p* = 0.0342), and adjuvant chemotherapy (HR for death: 0.62, 95% CI: 0.52–0.74, *p* < 0.0001) were significantly associated with longer OS. In multivariable analysis, younger age (*p* = 0.0101), female sex (*p* < 0.0001), race other than Caucasian (*p* = 0.0294), pathological N1 disease (*p* = 0.0173), and adjuvant chemotherapy (HR for death: 0.58, 95% CI: 0.48–0.71, *p* < 0.0001) were independent factors for predicting longer OS. After PSM for age, sex, histology, and pathological LN metastasis, multivariate analysis of OS showed that younger age (*p* = 0.0206), female sex (*p* = 0.0005), and adjuvant chemotherapy (HR for death: 0.58, 95% CI: 0.46–0.72, *p* < 0.0001) were independent factors for predicting longer OS (Table 3). The HRs for age, sex, and adjuvant chemotherapy were 0.76 (95% CI: 0.61–0.96), 0.66 (95% CI: 0.53–0.83), and 0.58 (95% CI: 0.46–0.72), respectively. The HR of adjuvant chemotherapy was the least among the three factors, suggesting that it was the most important clinical factor.

### 3.4. Subgroup Analyses of OS by Surgical Procedures in NSCLC Patients Who Underwent Sublobar Resection with Pathological LN Metastasis According to the Use of Adjuvant Chemotherapy

In the total cohort, 651 (80%) and 159 (20%) underwent wedge resection and segmentectomy, respectively. To find out if wedge resection and segmentectomy have the same tendencies with respect to the survival benefit of adjuvant chemotherapy, subgroup analyses according to surgical procedure were performed. In NSCLC patients who had wedge resection and pathological LN metastasis, adjuvant chemotherapy was significantly associated with longer OS than those without (median OS: 56.6 vs. 29.2 months, HR for death: 0.61, 95% CI: 0.50–0.75, *p* < 0.0001; Figure 4a). Similarly, in NSCLC patients who had segmentectomy and pathological LN metastasis, adjuvant chemotherapy was significantly associated with longer OS than in those without (median OS: 78.5 vs. 38.8 months, HR for death: 0.58, 95% CI: 0.39–0.87, *p* = 0.0081; Figure 4b).

## 4. Discussion

For patients with small NSCLC, lymph node metastasis is rare due to the early stage of the disease [12]. Therefore, the number of cases in which adjuvant therapy is indicated after surgery for patients with small NSCLC is small, making it difficult to study the efficacy of adjuvant therapy. Therefore, using the NCDB, a large database, we collected and analyzed rare cases from a long list of cases from 2004 to 2018. In this analysis, we selected patients who were considered to have no lymph node metastasis preoperatively (clinical N0) and were found to have lymph node metastasis postoperatively (pathological N1/N2). Until now, adjuvant chemotherapy for lymph node metastasis-positive cases after sublobar resection has been recommended to NSCLC patients and carried out in the absence of definitive evidence. Although this was a retrospective study and may be subject to selection bias, we are now better able to recommend and carry out adjuvant chemotherapy for these patients with these data.

In patients with NSCLC of 20 mm or smaller, hilar and mediastinal lymph node metastases occur in 10–15% of cases [12]. Such patients experience stage migration from stage I to stage II or III and an increased frequency of recurrence. In the current study, adjuvant chemotherapy was suggested to have a clinically significant survival benefit in patients with NSCLC ≤ 20 mm with pathologic LN metastases who underwent sublobar resection. As discussed above, since the application of surgical resection and postoperative adjuvant chemotherapy is influenced by patient age, comorbidities, and pathological N stage, only patients with a Charlson comorbidity score = 0 were included in this study to reduce such biases, and PSM was also performed. After adjusting age, sex, histology, and pathological N stage via PSM, adjuvant chemotherapy was an independent prognostic factor of longer OS, with a clinically meaningful low HR (0.58), and the 5-year survival rate was 16.4% higher with adjuvant chemotherapy than without. Moreover, the subgroup analysis by surgical procedure suggested a similar trend regarding the survival benefit of adjuvant chemotherapy regardless of wedge resection or segmentectomy.

There is no existing evidence on the effectiveness of adjuvant chemotherapy after sublobar resection of small-sized NSCLC with pathological LN metastasis. Recently, two major clinical trials on sublobar resection for small-sized NSCLC provided different protocols on when to use adjuvant chemotherapy. The CALGB 140503 study left the decision to perform adjuvant chemotherapy up to the physician’s choice [14]. In contrast, the JCOG0802/WJOG4607L trial generally recommended administering adjuvant chemotherapy for pathological stage II-III disease [15]. In the current real-world data, 567 out of 810 (70%) patients received adjuvant chemotherapy. Considering that our results suggested a survival benefit of adjuvant chemotherapy and that 30% of patients did not receive it, it may be important to consider expanding the indication for adjuvant chemotherapy to more patients with small-sized NSCLC and pathological LN metastasis after sublobar resection. The results of the current study also suggest the importance of intraoperative LN dissection to detect potential LN metastasis. Our previous retrospective study regarding the required extent of thoracic lymphadenectomy in patients with small-sized NSCLC who undergo sublobar resection reported that performing ≥ 10 LN dissections had a survival benefit [19].

Recently, adjuvant chemotherapy has dramatically developed in the area of targeted drugs and immune checkpoint inhibitors (ICI) [20,21,22]. The ADAURA trial on adjuvant osimertinib, a third-generation epidermal growth factor receptor (EGFR) tyrosine kinase inhibitor, has demonstrated a survival benefit [23]. Moreover, the ongoing ADAURA2 trial has been assessing the efficacy of adjuvant osimertinib in early-stage (IA2-IA3) NSCLC [24]. With regard to ICIs, the IMpower010 trial demonstrated significantly im-proved DFS with adjuvant atezolizumab versus best supportive care in programmed death-ligand 1-positive populations [17]. Although targeted drugs and ICI have been developed as adjuvant treatment, they were primarily investigated in the setting of post-adjuvant chemotherapy, except for the ALINA trial. Until future studies clearly determine the lack of OS benefit, the use of adjuvant chemotherapy prior to targeted drugs/ICI should be strongly considered in routine practice. Therefore, for patients with genetic mutations or programmed death-ligand 1 expression who undergo sublobar resection and have pathological LN metastases, considering platinum-based chemotherapy as an initial adjuvant treatment is advisable.

The current study had several limitations. First, it was a retrospective analysis subject to biases related to the surgeon’s decision making and patient characteristics such as performance status. Future research should aim to verify our results through well-planned randomized trials. However, the rarity of pathological LN metastasis in patients with small-sized NSCLC makes it challenging to conduct such prospective studies. Despite its retrospective nature, the NCDB provided a substantial dataset that facilitated our study [25,26]. Second, the current study lacked comprehensive data on recurrence or cause of death, which are crucial for assessing the clinical impact of adjuvant chemotherapy on cancer-specific outcomes. Third, the NCDB lacked the detailed data (station and number of metastases) on pathological LN metastasis. The IASLC Lung Cancer Staging Project proposed the addition of new sub-descriptors to N2 for single-station and multiple-station involvement based on the results that NSCLC patients with multiple-station N2 had a significantly shorter OS compared with those with single-station N2 (5-year survival rate: 38% vs. 49%) [27]. The survival benefits of adjuvant chemotherapy may differ depending on the station and number of pathological LN metastases, which should be investigated in further studies. Fourth, the NCDB lacks molecular data (i.e., *EGFR* and *ALK*). Since the prognostic impact of driver genes on survival is significant, they should be added to the survival analysis and require statistical processing [28,29].

## 5. Conclusions

Our retrospective study investigating patients with small-sized (≤20 mm) NSCLC with pathological LN metastasis who underwent sublobar resection elucidated that adjuvant chemotherapy was an independent factor of longer OS. Adjuvant chemotherapy may be recommended in patients with small-sized NSCLC with pathological LN metastasis even if the resection is sublobar. Further studies are warranted to validate these findings.

## Figures and Tables

**Figure 1 cancers-16-02176-f001:**
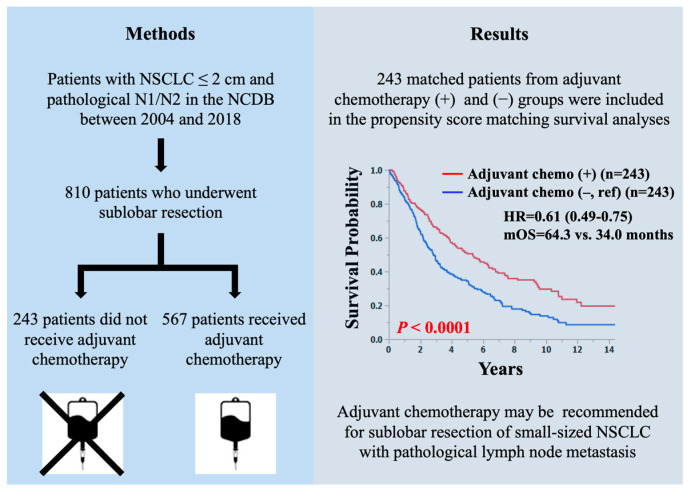
A graphical abstract of this study.

**Figure 2 cancers-16-02176-f002:**
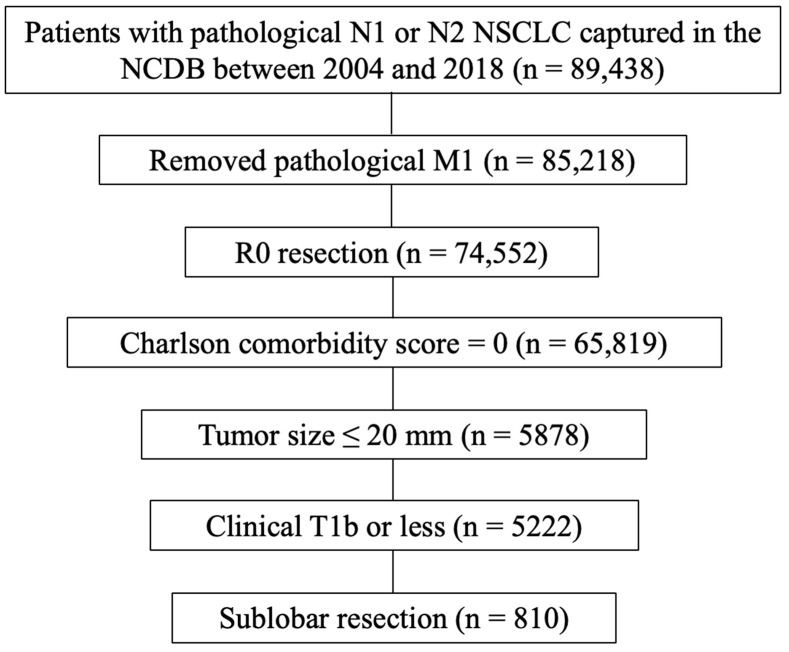
Study flow diagram of case eligibility. NSCLC, non-small-cell lung cancer; NCDB, National Cancer Database.

**Figure 3 cancers-16-02176-f003:**
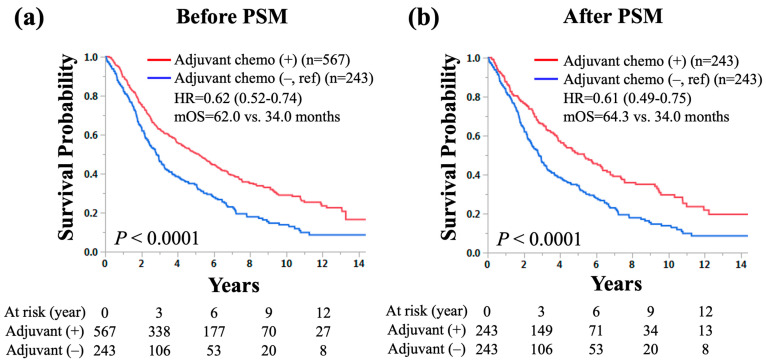
The Kaplan–Meier curves of overall survival according to the use of adjuvant chemotherapy in patients with small-sized (≤20 mm) non-small-cell lung cancer and lymph node metastasis who underwent sublobar resection (**a**) before and (**b**) after propensity score matching. HR: hazard ratio, mOS: median overall survival, PSM: propensity score matching.

**Figure 4 cancers-16-02176-f004:**
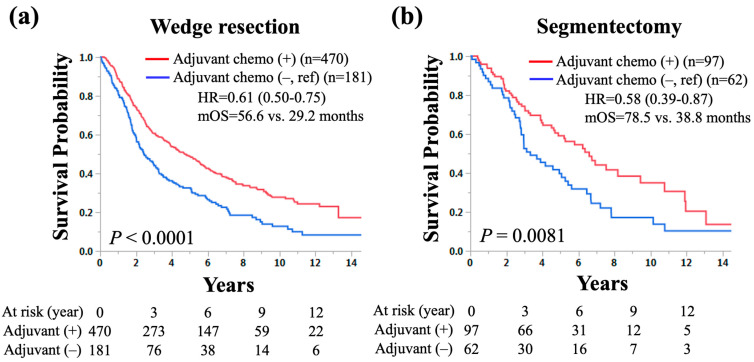
The Kaplan–Meier curves of overall survival according to the use of adjuvant chemotherapy in patients with small-sized (≤20 mm) non-small-cell lung cancer and lymph node metastasis who underwent (**a**) wedge resection and (**b**) segmentectomy. HR: hazard ratio, mOS: median overall survival.

**Table 1 cancers-16-02176-t001:** Clinical characteristics of patients with or without adjuvant chemotherapy who had sublobar resection of small-sized non-small-cell lung cancer ≤ 20 mm and pathological lymph node metastasis.

Factors		Before Propensity Score Matching (*n* = 810)			After Propensity Score Matching (*n* = 486)		
		Adjuvant Chemotherapy		*p* Value	Adjuvant Chemotherapy		*p* Value
		Present (*n* = 567)	Absent (*n* = 243)		Present (*n* = 243)	Absent (*n* = 243)	
Age	≥70	206 (36%)	132 (54%)	<0.0001	137 (56%)	132 (54%)	0.6482
	<70	361 (64%)	111 (46%)		106 (44%)	111 (46%)	
Sex	male	230 (41%)	109 (45%)	0.2565	110 (45%)	109 (45%)	0.9274
	female	337 (59%)	134 (55%)		133 (55%)	134 (55%)	
Race	Caucasian	490 (86%)	207 (85%)	0.6421	210 (86%)	207 (85%)	0.6966
	others	77 (14%)	36 (15%)		33 (14%)	36 (15%)	
Institution	academic	213 (38%)	123 (51%)	0.0006	98 (40%)	123 (51%)	0.0228
	others	354 (62%)	120 (49%)		145 (60%)	120 (49%)	
Year of diagnosis	2004–2009	211 (37%)	118 (49%)	0.0026	102 (42%)	118 (49%)	0.1448
	2010–2018	356 (63%)	125 (51%)		141 (58%)	125 (51%)	
Tumor size	≥10 mm	467 (82%)	194 (80%)	0.3948	191 (79%)	194 (80%)	0.7373
	<10 mm	100 (18%)	49 (20%)		52 (21%)	49 (20%)	
Laterality	right	287 (51%)	102 (42%)	0.0241	120 (49%)	102 (42%)	0.1012
	left	280 (49%)	141 (58%)		123 (51%)	141 (58%)	
Surgical procedure	wedge	470 (83%)	181 (74%)	0.0058	181 (74%)	181 (74%)	1.000
	segmentectomy	97 (17%)	62 (26%)		62 (26%)	62 (26%)	
Number of LNs dissected	≤9	405 (71%)	180 (74%)	0.5686	172 (71%)	180 (74%)	0.5254
	≥10	110 (20%)	46 (19%)		56 (23%)	46 (19%)	
	unknown	52 (9%)	17 (7%)		15 (6%)	17 (7%)	
Histology	squamous cell carcinoma	80 (14%)	40 (16%)	0.5350	45 (19%)	40 (16%)	0.8114
	adenocarcinoma	445 (79%)	182 (75%)		176 (72%)	182 (75%)	
	others	42 (7%)	21 (9%)		22 (9%)	21 (9%)	
Pathologic N stage	N2	401 (71%)	125 (51%)	<0.0001	130 (53%)	125 (51%)	0.6497
	N1	166 (29%)	118 (49%)		113 (47%)	118 (49%)	
Adjuvant chest radiation	yes	269 (47%)	26 (11%)	<0.0001	50 (21%)	26 (11%)	0.0027
	no	298 (53%)	217 (89%)		193 (79%)	217 (89%)	

**Table 2 cancers-16-02176-t002:** Univariate and multivariable analyses of overall survival in patients with non-small-cell lung cancer ≤ 20 mm who had sublobar resection and pathological lymph node metastases before propensity score matching (*n* = 810).

Factors		Univariate	Multivariable
		Hazard Ratio (95% CI), *p* Value	Hazard Ratio (95% CI), *p* Value
Age	<70	0.77 (0.65–0.91), 0.0026	0.79 (0.66–0.95), 0.0101
	≥70 (Ref)		
Sex	female	0.66 (0.56–0.79), <0.0001	0.67 (0.56–0.79), <0.0001
	male (Ref)		
Race	others	0.78 (0.60–1.01), 0.0503	0.76 (0.58–0.97), 0.0294
	Caucasian (Ref)		
Institution	academic	0.86 (0.72–1.02), 0.0772	0.86 (0.72–1.02), 0.0834
	others (Ref)		
Year of diagnosis	2010–2018	0.85 (0.72–1.01), 0.0714	0.89 (0.74–1.06), 0.1901
	2004–2009 (Ref)		
Tumor size	<10 mm	0.84 (0.67–1.05), 0.1276	0.84 (0.67–1.05), 0.1342
	≥10 mm (Ref)		
Laterality	right	0.95 (0.81–1.13), 0.5866	1.01 (0.85–1.21), 0.8701
	left (Ref)		
Surgical procedure	segmentectomy	0.93 (0.71–1.19), 0.5503	0.83 (0.66–1.04), 0.1071
	wedge (Ref)		
Number of LNs dissected	≥10	0.74 (0.58–0.92), 0.0073	0.80 (0.63–1.02), 0.0718
	unknown	0.88 (0.65–1.18), 0.4182	0.87 (0.64–1.19), 0.3893
	≤9 (Ref)		
Histology	adenocarcinoma	0.82 (0.65–1.05), 0.1174	0.85 (0.67–1.08), 0.1844
	others	1.25 (0.88–1.76), 0.2139	1.11 (0.78–1.58), 0.5569
	squamous cell carcinoma (Ref)		
Pathological N stage	N1	0.83 (0.69–0.99), 0.0342	0.79 (0.65–0.96), 0.0173
	N2 (Ref)		
Adjuvant chest radiation	yes	1.05 (0.88–1.25), 0.6020	1.22 (0.99–1.50), 0.0554
	no (Ref)		
Adjuvant chemotherapy	yes	0.62 (0.52–0.74), <0.0001	0.58 (0.48–0.71), <0.0001
	no (Ref)		

**Table 3 cancers-16-02176-t003:** Univariate and multivariable analyses of overall survival in patients with non-small-cell lung cancer ≤ 20 mm who had sublobar resection and pathological lymph node metastases after propensity score matching (*n* = 486).

Factors		Univariate	Multivariable
		Hazard Ratio (95% CI), *p* Value	Hazard Ratio (95% CI), *p* Value
Age	<70	0.79 (0.63–0.98), 0.0303	0.76 (0.61–0.96), 0.0206
	≥70 (Ref)		
Sex	female	0.71 (0.57–0.87), 0.0015	0.66 (0.53–0.83), 0.0005
	male (Ref)		
Race	others	0.84 (0.61–1.14), 0.2800	0.85 (0.62–1.18), 0.3337
	Caucasian (Ref)		
Institution	academic	0.88 (0.71–1.09), 0.2358	0.85 (0.68–1.06), 0.1583
	others (Ref)		
Year of diagnosis	2010–2018	0.96 (0.77–1.20), 0.7202	0.98 (0.79–1.23), 0.8873
	2004–2009 (Ref)		
Tumor size	<10 mm	0.85 (0.64–1.10), 0.2163	0.89 (0.68–1.17), 0.3983
	≥10 mm (Ref)		
Laterality	right	0.84 (0.67–1.04), 0.1049	0.90 (0.72–1.13), 0.3819
	left (Ref)		
Surgical procedure	segmentectomy	0.93 (0.71–1.19), 0.5503	0.86 (0.66–1.12), 0.2645
	wedge (Ref)		
Number of LNs dissected	≥10	0.78 (0.58–1.03), 0.0752	0.83 (0.61–1.10), 0.2005
	unknown	0.91 (0.58–1.36), 0.6673	0.80 (0.50–1.21), 0.2968
	≤9 (Ref)		
Histology	adenocarcinoma	0.90 (0.71–1.16), 0.4230	0.96 (0.75–1.23), 0.7216
	squamous cell carcinoma/others (Ref)		
Pathological N stage	N1	0.78 (0.63–0.97), 0.0248	0.84 (0.67–1.05), 0.1341
	N2 (Ref)		
Adjuvant chest radiation	yes	1.30 (0.98–1.70), 0.0726	1.35 (0.99–1.82), 0.0552
	no (Ref)		
Adjuvant chemotherapy	yes	0.61 (0.49–0.75), <0.0001	0.58 (0.46–0.72), <0.0001
	no (Ref)		

## Data Availability

The data that support the findings of this study are available from the corresponding author upon reasonable request.

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
