# Peer review of "The Clinical Role of Adjuvant Chemotherapy after Sublobar Resection for Non-Small-Cell Lung Cancer ≤ 20 mm with Lymph Node Metastases: A Propensity-Matched Analysis of the National Cancer Database"

_cancers, 2024, doi:10.3390/cancers16122176_

Round 1

Reviewer 1 Report

Comments and Suggestions for Authors

The study is very interesintg and accurateyl consctructed. Limitations are very well described.  Only minor issues are herein suggetsed.

1. Line 109. wedge resection (p = 0.0058) is reported to be significantly associated to adjuvant chemotherapy. What are the other surgical procedures ? Is it written in lines 155-156?

 2. Figure 2. After PSM (b), mOS changed from 62.0 vs 34.0 to 64.3 vs 34.0. Is it possible that, after the correction, median OS increases in the study group?

 3. It may useful to know some data on chemotherapy performed, if psosibile. Platinum-based? Not immune checkpoint inhibitors?

4. Some spelling mistakes. Line 157: teden-cies; Line 158: ac-cording...

Author Response

1) Thank you very much for your comments. As you pointed it out, the other surgical procedure, segmentectomy, should be mentioned as in lines 155-156. We have clarified this point in the manuscript to make it more explicit (line 128-129).

2) Thank you very much for your comments. We understand that it is questionable why median OS increased in the study group after PSM (Patient with poor conditions in the study group should be matched and results should be poorer). We think that the increase in median OS after PSM may be due to the imbalanced patient characteristics other than adjusted variables (age, sex, histology, and pathological N stage). Anyway, we believe that PSM analysis enhances the reliability of the survival estimates.

3) Thank you very much for your valuable suggestion. We agree that it is important to show the types of chemotherapy used, but there is no information on systemic treatment regimen, dose, schedule in the NCDB. Given that adjuvant atezolizumab and pembrolizumab were approved by the FDA in 2021 and 2023, respectively, the majority of the types of adjuvant chemotherapy used for NSCLC patients between 2004 and 2018 should have been platinum-based. Very few cases of immune checkpoint inhibitors, if any, were included that used in a clinical trial setting. We added this discussion to the manuscript (line 102-106).

4) Thank you very much for your pointing out these errors. We have corrected the spelling mistakes in the manuscript (lines 184, 185, 186 187, 189).

Reviewer 2 Report

Comments and Suggestions for Authors

Dear author, Thank you very much for reviewing your article. I give my suggestion, which you may address in your research manuscript.

How does sublobar resection compare to other surgical treatments for small-sized non-small cell lung cancer (NSCLC) in terms of patient outcomes? Considering its frequent use, what specific advantages does sublobar resection offer for small-sized NSCLC, and how do these benefits impact overall survival rates for patients?

What part does adjuvant chemotherapy play in the management of pathological lymph node metastases following sublobar resection in small-sized non-small cell lung cancer? In considering the lack of evidence about its efficacy, what potential effects on long-term survival outcomes may adjuvant chemotherapy have on these patients?

What insights were gained into the effectiveness of adjuvant treatment for patients with small-sized NSCLC who had sublobar resection thanks to the National Cancer Database? Which particular patient data between 2004 and 2018 was examined to ascertain how adjuvant chemotherapy affected survival?

What were the study's inclusion criteria, and why were they chosen in particular? What role did clinical stage, comorbidity score, and tumour size play in the patient population selection process for adjuvant chemotherapy trials?

What statistical methods were used to assess survival outcomes in the study, and how did these methods ensure the reliability of the results? How did the Kaplan-Meier method, log-rank test, and multivariable Cox proportional hazards analyses contribute to the study's findings?

Which statistical techniques were applied to evaluate the study's survival outcomes, and how were the results' dependability guaranteed by these techniques? What impact did the multivariate Cox proportional hazards analysis, log-rank test, and Kaplan-Meier technique have on the study's conclusions?

What were the study's primary conclusions about the benefits of adjuvant chemotherapy for survival? What statistical significance was noted in the differences in median survival rates between individuals who got adjuvant chemotherapy and those who did not?

Which independent predictors of prolonged survival following PSM were found in the multivariate analyses? In what ways did the patient cohort's age, sex, and adjuvant chemotherapy independently impact survival outcomes?

What implications does the study's result on adjuvant chemotherapy's prognostic effect have for clinical practice treating small-sized NSCLC with pathological lymph node metastases? What effects may these results have on treatment plans and decision-making procedures in the future?

What possible weaknesses does the study have, and how might they be addressed in future research? What more research is required to confirm and build upon these findings, given the study's dependence on database data and statistical adjustments?

Best Regard

Author Response

1) Thank you for your comments. Sublobar resection for small NSCLC has been shown to preserve lung function and prolong overall survival compared to standard treatment lobectomy. Lung-sparing surgical procedures increase feasibility of post-recurrent therapy, resulting in longer survival. Although this is an important point, the aim of the current study was to elucidate clinical impact of adjuvant chemotherapy in this population, not surgical procedure itself. Therefore, no amendments were made at this time, but this is considered an issue for future consideration.

2) Thank you for your comments. We performed this study given the lack of evidence about efficacy of adjuvant chemotherapy after intentional sublobar resection. The current study suggested that adjuvant chemotherapy prolonged overall survival in patients with pathological lymph node metastases following sublobar resection in small-sized NSCLC. As you pointed out, the potential effects of adjuvant chemotherapy on long-term survival remain unclear. Although we believe primary benefit from adjuvant chemotherapy for node-positive disease is to prevent distant metastasis, and local relapse may also be prevented modestly, further studies are needed to elucidate such potential effect.

3) Thank you for your comments. For patients with small NSCLC, lymph node metastasis is rare due to the early stage of the disease. Therefore, the number of cases in which adjuvant therapy is indicated after surgery for patients with small NSCLC is small, making it difficult to study the efficacy of adjuvant therapy. Therefore, using the National Cancer Database, a large database, we collected and analyzed rare cases from a long list of cases from 2004 to 2018. In this analysis, we selected patients who were considered to have no lymph node metastasis preoperatively (clinical N0) and were found to have lymph node metastasis postoperatively (pathological N1/N2). This analysis revealed that adjuvant chemotherapy significantly prolonged survival. We added these comments to the discussion part of the manuscript (lines 199-210).

4) Thank you for your comments. Historically, sublobar resection has been performed as a 'non-intentional' surgery primarily for patients who are unable to undergo lobectomy due to their comorbidity, low pulmonary function, and/or old age. However, there is a growing trend to identify even earlier lesions within ≤ 2 cm without lymph node metastasis and to apply 'intentional sublobar resection' to patients who are eligible for lobectomy. The purpose of our study was to investigate survival benefit of adjuvant chemotherapy in patients with small-sized (≤ 2 cm) NSCLC who underwent 'intentional sublobar resection', even if the patients were eligible for lobectomy. Therefore, we selected clinical stage IB or less, comorbidity score = 0, and tumor size ≤ 2 cm to exclude 'non-intentional sublobar resection'. We added these comments to the introduction and methods parts (lines 91-95).

5) We appreciate the reviewer’s comments. The primary outcome variable in this manuscript is overall survival (OS), which is a time-to-event type of variable. It contains combined information of whether or not an event (death) occurred, and when that event occurred. Traditional statistical methods such as logistic and linear regression are not suitable to handle both the event and time aspects of the outcome in the model. Also, traditional regression methods cannot handle the situation of censoring – a special type of missing data that the subjects did not experience the event of interest during the follow-up time. Statistically there are three main approaches to analyzing time-to-event data: non-parametric approach, semi-parametric approach, and parametric approach. We used non-parametric and semi-parametric approaches in this paper.

              The Kaplan-Meier estimator that we used is the most common non-parametric statistical method to handle time-to-event data. It works by breaking up the estimation of survival function S(t) into a series of steps (or intervals) based on the observed event times. Using Kaplan-Meier method, the estimated survival function can be plotted as a stepwise function with time on the X-axis (as can be seen in Figure 3 and 4 in our modified manuscript). Rank-based tests are non-parametric tests that can be used to statistically test the difference between survival curves. One of the most commonly used rank-based tests seen in the literature is the log-rank test that we used in this paper.

              The non-parametric approach to the time-to-event data is used to simply describe the survival function with respect to one factor (usually the main grouping variable) – referred as univariable model. However, the semi-parametric and fully-parametric models allows us to investigate the time relationship between time-to-event outcome and multiple factors at the seme time.

The Cox-proportional hazard regression model is the most commonly used semi-parametric statistical approach. It is a time-to-event regression model, which describes the relationship between the hazard function h(t) with a set of covariates. It is considered as a semi-parametric approach because the model has a non-parametric component (the baseline hazard) and a parametric component (the covariate vector). The Cox-proportional hazard regression will allow us to evaluate each potential contribution factor individually (in the univariate model), or jointly (in the multivariate model). The effect of each factor on the time-to-event outcome is quantified by hazard ratios (HR) and their 95% confidence intervals. If a confidence interval of the HR does not contain 1, it indicates that the associated factor’s contribution to the time-to-event outcome is statistically significant. We have the univariate and multivariate Cox-proportional hazard regression results listed in our Table 2 and Table 3.

              We hope the above descriptions answer the reviewer’s question. A comprehensive review of the statistical methods used for survival outcomes can be found in the following website: https://www.publichealth.columbia.edu/research/population-health-methods/time-event-data-analysis.

Reference for the above answer:

  • Goel MK, Khanna P, Kishore J. Understanding survival analysis: Kaplan-Meier estimate. Int J Ayurveda Res. 2010 Oct;1(4):274-8. doi: 10.4103/0974-7788.76794. PMID: 21455458; PMCID: PMC3059453.
  • Abd ElHafeez S, D'Arrigo G, Leonardis D, Fusaro M, Tripepi G, Roumeliotis S. Methods to Analyze Time-to-Event Data: The Cox Regression Analysis. Oxid Med Cell Longev. 2021 Nov 30;2021:1302811. doi: 10.1155/2021/1302811. PMID: 34887996; PMCID: PMC8651375.

6) We appreciate the reviewer’s comments. Please refer to our answers above for the description of the statistical methods we used to analyze the main outcome OS, a time-to-event variable. For the second part of your question, the Kaplan-Meier methods were used to estimate and plot the survival function S(t) against time. The non-parametric log-rank test was used to compare the survival function against our main categorical factor under investigation (the use of adjuvant chemotherapy). The results are shown in Figure 2 and Figure 3 (now revised as Figure 3 and Figure 4). Then semi-parametric Cox-proportional hazard regression models were used to evaluate the effect of each individual factor separately (in the univariate Cox model) and jointly (in the multivariate Cox model). The results are shown in our Table 2 and Table 3. Please note that the above statistical approach is very standard for handling the time-to-event data and is widely adopted in medical scientific research.

7) Thank you for your comments. 5-year survival rates of patients with adjuvant chemotherapy and those without after PSM were 50.6% and 34.2%, respectively. Thus, 5-year survival benefit by adjuvant chemotherapy was 16.4%. Additionally, as shown in Figure 2b (now revised as Figure 3b), their survival difference was statistically significant (P < 0.0001). We added some comments to the manuscript (lines 149-150).

8) Thank you for your comments. Independent predictors of prolonged survival following PSM in the multivariate analyses are age, sex, and adjuvant chemotherapy, shown in Table 3 and described in lines 166-169. The hazard ratios for age, sex and adjuvant chemotherapy were 0.76 (95% CI: 0.61-0.96), 0.66 (95% CI: 0.53-0.83), and 0.58 (95% CI: 0.46-0.72), respectively. The hazard ratio of adjuvant chemotherapy was the least among the 3 factors, suggesting that it was the most important clinical factor. We added these results to the manuscript (line 169-172). Aging primarily determines the overall survival, and male sex is also historically reported to be independent factor of OS, probably due to shorter life span than female, and likelihood of smoking. Regarding adjuvant chemotherapy, we described the discussion in Reviewer 2, comment 2.

9) Thank you for your comments. Until now, adjuvants chemotherapy for lymph node metastasis-positive cases after sublobar resection has been recommended to NSCLC patients and carried out in the absence of definitive evidence. Although this was a retrospective study and may be subject to selection bias, we are now better able to recommend and carry out adjuvant chemotherapy for these patients with this data. We added these comments to the manuscript (lines 206-210).

10) Thank you for your comments. Regarding weakness, this study looked back at past cases, which means it might have biases based on the surgeon's choices and the patients' conditions. Future studies should confirm our findings with carefully designed random trials. However, it's hard to do these studies because it's rare to find patients with small lung cancer who have lymph node metastasis. We have already addressed this issue (lines 256-261), so we did not add comments to the manuscript.

Reviewer 3 Report

Comments and Suggestions for Authors

1)      I strongly advise the authors to draw a graphical abstract at the end of the introduction. This helps the audience to understand your work more easily.

2)      The authors did not describe their statistical analysis step–by–step. They just referred to the figure 1.

3)      The authors must describe what is adjuvant chemotherapy, and which anti-cancer can be combined with an adjuvant, and discuss the immune mechanisms of using the adjuvant in the chemotherapy of patients very well.

4)      There is no description after the table/figure in the result part.

5)      The authors should conclude how much was effective using the adjuvant in terms of prolonging the survival rate compared to the non-adjuvant. 

Author Response

1) Thank you for your suggestion. We have added a graphical abstract at the end of the introduction to help the audience understand our work more easily (Figure 1, lines 78-79).

2) We appreciate the reviewer’s comments. Our Figure 1 (now revised as Figure 2) is a flow chart describing the selection process of our study sample. This does not overlap with our statistical methods, which is described specifically in subsection 2.2 (statistical analysis). A step-by-step statistical analytical flow is described as follows: (1) summarization of patients’ characteristics and comparisons between cohorts with and without adjuvant chemotherapy (Table 1); (2) Kaplan-Meier analysis of the OS between adjuvant chemotherapy groups (Figure 2 and Figure 3); and (3) Univariate and multivariate Cox-proportional hazard regression analysis (Table 2 and Table 3). In all the above three steps, the results were presented before and after the propensity score matching (PSM). The description of the statistical analytical flow is consistent with the order described in our statistical methodology subsection (section 2.1). To avoid confusion, our Figure 2 (flow chart) was moved from right below the statistical methods section to the end of section 2.1 instead (lines 109-110).

3) Thank you for your comments. We agree that it is important to show what is adjuvant chemotherapy, but there is no information on systemic treatment regimen, dose, schedule in the NCDB. Given that adjuvant atezolizumab and pembrolizumab were approved by the FDA in 2021 and 2023, respectively, the majority of the types of adjuvant chemotherapy used for NSCLC patients between 2004 and 2018 should have been platinum-based. Very few cases of immune checkpoint inhibitors, if any, were included that used in a clinical trial setting. We added this discussion to the manuscript (line 102-106). Regarding which anti-cancer can be combined with adjuvant chemotherapy, and the immune mechanisms of using adjuvant chemotherapy, we can only speculate them and can’t discuss based on our results. Therefore, although they are very important points, we did not add such discussion to the manuscript. Future studies focusing on these points should be performed.

4) Thank you for your comment. There is description before the table/figure in the result part. We referred to other articles published in Cancers and found that description before the table/figure in the result part was common. Therefore, we did not modify the manuscript.

5) Thank you for your helpful suggestion. We agree that we should conclude how much was effective using the adjuvant in terms of prolonging the survival rate compared to the non-adjuvant. As described in lines 147-149, 5-year survival rates of patients with adjuvant chemotherapy and those without after PSM were 50.6% and 34.2%, respectively. Thus, 5-year survival benefit by adjuvant chemotherapy was 16.4%. Additionally, as shown in Figure 2b (now revised as Figure 3b), their survival difference was statistically significant (P < 0.0001). We added some comments to the manuscript (lines 149-150).

Round 2

Reviewer 3 Report

Comments and Suggestions for Authors

The authors answered my comments, and I recommend publishing.